# Dietary Protein Quality Affects the Interplay between Gut Microbiota and Host Performance in Nile Tilapia

**DOI:** 10.3390/ani14050714

**Published:** 2024-02-24

**Authors:** Gabriella do Vale Pereira, Carla Teixeira, José Couto, Jorge Dias, Paulo Rema, Ana Teresa Gonçalves

**Affiliations:** 1SPAROS Lda., 8700-221 Olhão, Portugal; jorgedias@sparos.pt; 2Aquaculture Department, Center of Agrarian Sciences, Federal University of Santa Catarina (UFSC), Florianópolis 88061-600, Brazil; 3Riasearch Lda., 3870-168 Murtosa, Portugal; carlateixeira@riasearch.pt (C.T.);; 4Departamento de Zootécnia, Universidade de Trás os Montes e Alto Douro (UTAD), 5000-801 Vila Real, Portugal; prema@utad.pt; 5GreenCoLab, Associação Oceano Verde, 8500-139 Faro, Portugal

**Keywords:** protein sources, gut health, microbiome, *Oreochromis niloticus*, amino acid intake

## Abstract

**Simple Summary:**

Gut health depends on a complex network interaction between the host and the mucosal-associated microbiota, and it is modulated by dietary inputs. To gain knowledge on this interaction and shed light on the nutritional impacts at this level, this study assessed the effect of dietary protein quality on the modulation of different gut health parameters. Fish performance and gut microbiome abundance were correlated with amino acid intake and the functional prediction of bacteria metabolism. The results indicate that different protein qualities modulate the relationship between bacteria functional pathways and amino acid intake.

**Abstract:**

Dietary protein quality plays a key role in maintaining intestinal mucosal integrity, but also modulates the growth of luminal microorganisms. This work assessed the effect of dietary protein sources on the performance, gut morphology, and microbiome in Nile tilapia. Four isonitrogenous and isolipidic diets comprising equivalent amounts of the protein supply derived from either PLANT, ANIMAL, INSECT, or BACTERIAL (bacterial biomass) sources were fed to triplicate groups of fish (IBW: 12 g) for 46 days. Fish fed the ANIMAL and BACTERIAL diets showed significantly higher weight gains than those fed the PLANT and INSECT diets (*p* < 0.05). Relative abundance at the phylum level showed that Bacteroidetes, Fusobacteria, and Proteobacteria were the more abundant phyla in tilapia’s intestine, while *Cetobacterium* was the most representative genus in all treatments. Interesting patterns were observed in the correlation between amino acid intake and genus and species abundance. Metabolism prediction analysis showed that BACTERIAL amine and polyamine degradation pathways are modulated depending on diets. In conclusion, different protein sources modulate the relationship between bacteria functional pathways and amino acid intake.

## 1. Introduction

Meeting the ever-increasing demand for protein, within environmental limits, is one of the biggest challenges faced by the global food system in the 21st century. In a resource-constrained world, aquaculture is an attractive option for expanding the animal protein supply. However, the current trend towards sustainable growth and the eco-intensification of aquaculture have brought new challenges to the aquafeed industry. There is a need to use novel sources of proteins and oils and increase the incorporation of improved industrial by-products, functional ingredients, and additives to promote the welfare and health resistance of fish as well as mitigate the environmental impact of feeding.

Gut health affects animal performance, feed efficiency, and overall health and welfare; thus, its maintenance is key for animal nutrition. Gaining increasing attention in aquaculture, it depends on a complex balance of diet, mucosa integrity, the immune system, and commensal microbiota. As one of the main bulk ingredients in fish feeds, dietary protein supply (i.e., amino acids) not only plays a core role in fish development but is also pivotal for the maintenance of the intestinal mucosal integrity, while providing nutritional support to gut microorganisms in the gut [1,2,3,4]. In a recent examination [5], the functions and mechanisms of action of amino acids on the intestinal physiology and health of land-dwelling vertebrates were delineated. Certain amino acids, such as glutamate, glutamine, and aspartate, are catabolized by enterocytes to fulfill their increased energy demands and produce bioactive metabolites like glutathione or nitric oxide. In contrast, other amino acids (such as threonine and tryptophan) are primarily employed as essential components for protein synthesis in connection with mechanisms for epithelium renewal and mucin production. Several amino acids have been demonstrated to support the endocrine and barrier function of the intestine. Furthermore, amino acids are also metabolized by the gut microbiota that employ them for protein synthesis and in catabolic reactions releasing metabolites into the intestinal lumen, such as ammonia, hydrogen sulfide, branched-chain amino acids, polyamines, and phenolic compounds. Some of these, like hydrogen sulfide, a bacterial metabolite derived from cysteine, can affect epithelial energy metabolism and induce mucosal inflammation when present in excess, whereas others (like indole derivatives originating from tryptophan) tend to prevent dysfunction of the gut barrier or regulate enteroendocrine functions. The investigation into how fish gut microbiota make use of accessible amino acids is still ongoing. Nevertheless, in different species like pigs, it is established that amino acids in the intestine serve as the primary components of bacterial protein in the ileum [6,7] and also that amino acid intake varies depending on the BACTERIAL abundance [8].

In the year 2017, tilapia obtained the fourth position when considering both production quantity and value within the aquaculture species group [9]. The leading species of cultured tilapia is the Nile tilapia (*Oreochromis niloticus*). The grow-out feeds for tilapia generally contain moderate levels of crude protein (ranging from 28% to 36%), which are mainly derived from defatted oilseed meals like soybean, peanut, and rapeseed. Additionally, these feeds consist of relatively low levels (ranging from 3% to 10%) of fishmeal and/or land-animal processed by-product meals such as poultry meal. Considering the large scale of tilapia production, even a slight increase in production volumes would result in a significant rise in the demand for protein sources for their feeds. Recent studies have explored the use of emerging protein sources in Nile tilapia feeds, including insects [10], seaweeds [11], microalgae [12], yeasts [13], bacteria [14], and land-animal processed by-product meals [15]. These are some of the alternative protein sources that have been suggested to reduce reliance on fishmeal and promote tilapia growth. However, only a few studies have investigated their effects on intestinal health criteria, such as microbiota and mucosa integrity. Therefore, the objective of this trial was to compare the impact of different dietary protein qualities on gut morphology, microbiome, and amino acid intake in Nile tilapia juveniles.

## 2. Materials and Methods

### 2.1. Experimental Diets

The trial comprised four dietary treatments, isonitrogenous (35% crude protein), isolipidic (9% crude fat), and isoenergetic (gross energy 18.8 MJ/kg). The main variable among the diets was the nature of the 30% the total protein supply, which was derived from either fishmeal and poultry meal (ANIMAL), dehulled solvent-extracted soybean meal and corn gluten meal (PLANT), BACTERIAL biomasses from Corybacterium glutamicum and Methylococcus capsulatus (BACTERIAL), or a defatted Tenebrio molitor larvae meal (INSECT) (Table 1). The remaining 70% of the total protein supply was derived from corn gluten meal, dehulled solvent-extracted soybean meal, rapeseed meal, wheat grain, full-fat rice bran, wheat, and corn meal, which were kept fairly constant among the various diets. Fish oil and soybean oil were used as the main lipid sources, with a slight increase in fish oil in the fishmeal-free diets to guarantee constant levels of eicosapentaenoic acid (EPA) and docosahexaenoic acid (DHA). All diets were supplemented with selected crystalline indispensable amino acids and monocalcium phosphate to avoid any essential amino acid or phosphorus imbalance. Diets also comprised 1% chromium oxide as an inert digestibility marker.

Diets were manufactured by extrusion at the SPAROS feed mill (Olhão, Portugal). All powder ingredients were mixed according to the target formulation in a double-helix mixer (model 500 L, TGC Extrusion, Roullet-Saint-Estèphe, France) and ground (below 400 µm) in a micropulverizer hammer mill (model SH1, Hosokawa-Alpine, Augsburg, Germany). Diets (pellet size: 2.0 mm) were manufactured with a twin-screw extruder (model BC45, Clextral, Firminy, France) with a screw diameter of 55.5 mm. Extrusion conditions: feeder rate (73–76 kg/h), screw speed (246–252 rpm), water addition (345 mL/min), temperature barrel 1 (30–32 °C), temperature barrel 3 (111–114 °C). Extruded pellets were dried in a vibrating fluid bed dryer (model DR100, TGC Extrusion, Firminy, France). After cooling, oils were added by vacuum coating (model PG-10VCLAB, Dinnissen, Sevenum, The Netherlands). Coating conditions: pressure (700 mbar); spraying time under vacuum (approximately 90 s); return to atmospheric pressure (120 s). Immediately after coating, the diets were packed in sealed plastic buckets and shipped to the research site where they were stored at room temperature, but in a cool and aerated emplacement. Representative samples of each diet were taken for analysis.

The amino acid intakes of the experimental diets, expressed as a percentage relative to the amino acid intake of the ANIMAL diet, are presented in Figure 1 (absolute amino acid values are presented in Appendix A Appendix A). Although formulated to cover the known amino acid requirements of Nile tilapia, the dietary changes altered the amino acid profiles among diets. In comparison to the ANIMAL diet, some of the most striking differences worth mentioning were the lower contents of Arg, His, and Gly in the PLANT, BACTERIAL, and INSECT diets; the higher content of Thr found in the BACTERIAL diet; the lower contents of Cys in the PLANT and BACTERIAL diets; and the higher levels of Glx (glutamic acid + glutamine) in the PLANT and INSECT diets.

### 2.2. Growth Trial

The trial was conducted at the experimental facilities of the University of Trás-os-Montes e Alto Douro (UTAD, Vila Real, Portugal), which are registered for experimentation with aquatic species (0421/2018) by Direção-Geral de Alimentação e Veterinária (Ministry of Agriculture, Lisboa, Portugal). The experimental protocol was approved by the Animal Welfare Committee—Órgão Responsável pelo Bem-Estar Animal (ORBEA) of UTAD and was performed in compliance with the European (Directive 2010/63/EU) and Portuguese (Decreto-Lei nº. 113/2013, August 7th) legislation on the protection of animals for scientific purposes.

Nile tilapia (*Oreochromis niloticus*) originated from TIL-Aqua International (Velden, The Netherlands). Triplicate groups of 15 fish with a mean initial body weight (IBW) of 12.1 ± 1.3 g were fed one of the four experimental diets over 46 days. Fish were grown in glass rectangular aquaria (volume: 90 L) supplied with recirculated freshwater (water flow rate: 4.5 L/min). The average water temperature during the trial was 24.8 ± 0.3 °C and the dissolved oxygen levels were kept above 6.5 mg/L. The fish were subjected to a photoperiod regime of 16 h light and 8 h dark. Fish were hand-fed to visual satiety, in 3 meals per day (09:00, 14:00, and 17.00 h). Utmost care was taken to avoid feed wastage and allow for precise quantification of feed intake. Allocation of the test diets to the experimental units (tanks) was made by full randomization. After being subjected to lethal anesthesia (250 mg/L ethylene glycol monophenyl ether, Sigma-Aldrich, Salamanca, Spain), a pool of 10 whole fish from the initial stock (start of the trial) and a pool of 5 whole fish from each replicate tank at the end of the trial were sampled and stored at −20 °C for subsequent analysis of whole-body composition. For the microbiome screening, three fish from each replicate had their peritoneal area cleaned with 70% ethanol to avoid bacterial contaminant, and a cut from the anus to the intraperitoneal cavity was made to remove the intestine, which was sampled and placed on a sterile Petri dish. The posterior intestines from the fish were sampled and the intestinal contents were removed using forceps. The intestinal tissue (mucosa) was washed with sterile PBS to remove any digest attached to the intestine walls and then placed into DNA-free cryovials. The samples were stored at −80 °C until further analysis. Sections of approximately 1 cm of the hindgut of the same 3 fish that were sampled for their microbiome were dissected and placed immediately in 4% formaldehyde for further histological analysis.

### 2.3. Apparent Digestibility Measurements

In parallel to the growth performance trial, groups of 15 fish from the same initial stock, with a mean body weight (BW) of 25.3 ± 3.1 g, were used to determine the apparent digestibility coefficients (ADC) of the dietary components by the indirect method with diets containing 1% chromium oxide as the inert tracer. Each dietary treatment was tested in triplicate. Fish were stocked in cylindro-conical tanks (volume: 60 L) and supplied with recirculated freshwater at an average water temperature of 25 °C. Before the start of feces collection, fish were adapted over 16 days to the experimental diets. Fish were fed twice a day (9.00 and 14.00 h) by hand in slight excess. After the last daily meal, tanks were thoroughly cleaned to eliminate any feed residues, and feces were collected daily for the following 12 days using a mechanical system for the continuous filtration of feces in the outlet water (INRA system). After daily collection, the feces from each replicate tank were frozen at −20 °C and subsequently freeze-dried before analysis.

### 2.4. Analytical Methods

#### 2.4.1. Nutritional Characterization

Analysis of diets, whole fish, and feces followed the methodology described by AOAC [16]. Dry matter after drying at 105 °C for 24 h; total ash by combustion (550 °C during 6 h) in a muffle furnace (Nabertherm L9/11/B170, Lilienthal, Germany); crude protein (Nx6.25) by a flash combustion technique, followed by a gas chromatographic separation and thermal conductivity detection with a Leco N Analyzer (Model FP-528, Leco Corporation, Benton Harbor, MI, USA); crude lipid by petroleum ether extraction (40–60 °C) using a Soxtec™ 2055 Fat Extraction System (Foss, Hillerød, Denmark), with prior acid hydrolysis with 8.3 M HCl; gross energy in an adiabatic bomb calorimeter (Werke C2000, IKA, Hohenems, Germany). Chromium concentrations in feeds and feces were determined according to Bolin et al. [17], after perchloric acid digestion. Amino acids were determined after hydrolysis in 6M HCL at 108 °C for 24 h in nitrogen-flushed glass vials. A Waters Pico-Tag reversed-phase HPLC system using norleucine as an internal standard was used. The resulting chromatograms were analyzed with Breeze software (Waters, Milford, MA, USA). The tryptophan in complete feeds was analyzed according to ISO 13904:2016 by HPLC-FD methodology [18].

#### 2.4.2. Microbiome

DNA extraction

The DNA was extracted individually from a total of 36 samples (3 per replicate) using the DNA extraction High Pure PCR Template Preparation Kit (Roche, Amadora, Portugal). A lysozyme pre-lysis step was performed on each of the samples to guarantee the extraction of Gram-positive bacteria by adding 500 µL of freshly made lysozyme (50 mg/mL of TE buffer or PBS) per sample and incubating for 30 min at 37 °C. The following DNA extraction procedure was conducted according the manufacturer’s protocol. The extracted DNA from all 36 samples was sent to the Genoinseq (Cantanhede, Portugal) laboratory for generation of the raw sequence data of the 16S V1 and V2 amplicon through next-generation sequencing of DNA molecules.

High-throughput sequencing

Samples were prepared for Illumina Sequencing by 16S rRNA gene amplification of the BACTERIAL community. The DNA was amplified for the hypervariable V1–V2 region with specific primers and further reamplified in a limited-cycle PCR reaction to add sequencing adapters and dual indexes. The first PCR reactions were performed for each sample using a KAPA HiFi HotStart PCR Kit according to manufacturer suggestions, with 0.3 μM of each PCR primer: forward primer 27F 5′–AGAGTTTGATCMTGGCTCAG-3′ and a pool of reverse primers: 338R-I-R 5′–GCWGCCTCCCGTAGGAGT-3′ and 338R-II-R 5′–GCWGCCACCCGTAGGTGT-3′ [19] as well as 12.5 ng of template DNA in a total volume of 25 μL. The PCR conditions involved a 3 min denaturation at 95 °C, followed by 35 cycles of 98 °C for 20 s, 56 °C for 30 s, and 72 °C for 30 s, and a final extension at 72 °C for 5 min. The second PCR reactions added indexes and sequencing adapters to both ends of the amplified target region according to the manufacturer’s recommendations [20]. Negative PCR controls were included for all amplification procedures. PCR products were then one-step purified and normalized using a SequalPrep 96-well plate kit (ThermoFisher Scientific, Waltham, MA, USA) [21], and pooled and paired-end (2 × 300 bp) sequenced in the Illumina MiSeq^®^ sequencer with the V3 chemistry, according to manufacturer’s instructions (Illumina, San Diego, CA, USA) at Genoinseq (Cantanhede, Portugal). Sequence data were processed at Genoinseq (Cantanhede, Portugal). Raw reads were extracted from the Illumina MiSeq^®^ System in fastq format for further analysis.

Bioinformatics

Sequences were demultiplexed, and non-biological nucleotides (e.g., primers, adapters) were removed according to the in-house protocols of the sequencing provider. Bioinformatic analyses were performed with QIIME 2 VERSION 2021.11 [22]. Paired-end raw sequences were trimmed and truncated to perform a quality filter, after which they were merged and chimeras were removed by denoising with DADA2 via q2-dada2 script [23]. The alignment of the obtained amplicon sequence variants (ASVs) was performed with mafft [24], and fasttree2 (via q2-phylogeny) was used to construct the phylogeny [25]. Taxonomy was assigned to ASVs using the q2-feature-classifier [26] using the Greengenes 13_8 99% OTUS sequences database as reference [27]. To evaluate BACTERIAL diversity, samples were rarefied to 293 sequences per sample. The alpha diversity metrics Shannon, observed species, and faith_pd; and beta-diversity metrics bray_curtis_distance, weighted and unweighted unifrac were estimated using the q2-diversity script. Beta diversity results were further analyzed through principal coordinate analysis (PCoA) to represent inter-sample phylogenetic differences in a low-dimensional Euclidean space, using the q2-diversity script as well. Differences between the group’s alpha diversity metrics were assessed using QIIME2 significance tests based on the Kruskal–Wallis test, whereas the group’s beta diversities’ significance was tested by using PERMANOVA, which uses the distances between samples of the same group and compares them to the distances between groups [28,29]. Differential relative abundances at the genus level were assessed with DESeq2 [30].

The functional profile of the microbiome was predicted by using the Phylogenetic Investigation of Communities by Reconstruction of Unobserved States (PICRUST2) [31]. PICRUST2 considers that phylogenetically related organisms are more likely to have similar gene contents, and based on that concept the algorithm uses several gene family databases. Here, the functional profiles of the BACTERIAL communities from fish that were fed the four different diets were predicted using the MetaCyc pathways database [32]. The evaluation of the pathway’s proportions between the BACTERIAL communities from the intestines of Nile tilapia fed the different diets was performed using a statistical analysis of metagenomics profiles [33], considering an FDR-corrected *p*-value < 0.05 for acceptance of significant differences.

#### 2.4.3. Histology

Formalin-fixed hindgut samples were sent to Ictiovet (Spain) for semi-quantitative histopathological analysis and complementary morphometric analysis. Samples were distributed into 4 different diet groups containing different protein sources (ANIMAL, BACTERIAL, INSECT, and PLANT protein). The 4 μm thick sections were stained using haematoxylin and eosin for histopathological analysis. Semi-quantitative analysis of the most significant histopathological features of the intestine samples was performed according to the following criterion, including half scores for intermediate intensities: 0 = no incidence; 1 = mild incidence; 2 = moderate incidence; 3 = marked incidence. For morphometric analysis of the mucus cells, sections of 4 μm in thickness were cut. Sections were stained using periodic acid—Schiff (PAS)—Alcian Blue staining, a histochemical stain that stains mucins. Slides were digitalized using a Hammamatsu NanoZoomer 2.0 RS. Each section was analyzed for listed mucus cell parameters using ImageJ 1.52, (National Institutes of Health, Bethesda, MD, USA).

Additionally, image analysis software was used on hindgut sections to measure gut health indicators including internal (lumen) diameter (M)/external diameter (S) ratio, average intestinal fold height and width (average of 3 consecutive intestinal folds), as well as mucus cell counts.

### 2.5. Zootechnical Performance Evaluation Criteria

To evaluate growth performance, the following equations were used.
Weight gain (g): FBW − IBW(1)
Feed conversion ratio, FCR: crude feed intake/weight gain.(2)
Feed intake, FI (%BW/day): (crude feed intake/(IBW + FBW)/2/days) × 100.(3)
Protein efficiency ratio, PER: wet weight gain/crude protein intake.(4)
(5)$$\mathrm{Retention}\;\left(\%\right)=100\;\times \;\frac{\left(\mathrm{FBW}\;\times \;\mathrm{NFF}\right)\;-\;(\mathrm{IBW}\;\times \;\mathrm{NIF})}{\mathrm{Nutrient}\;\mathrm{intake}}$$
where: IBW (g): Initial mean body weight, FBW (g): Final mean body weight, NFF: Nutrient content of final fish, NIF: Nutrient content of initial fish.

Apparent digestibility coefficients (ADC) of dietary nutrients and energy in the experimental diets were calculated according to NRC (2011):(6)$$\mathrm{ADC},\;\%=100\;\times \;\frac{\%\;\mathrm{marker}\;\mathrm{diet}}{\%\;\mathrm{marker}\;\mathrm{feces}}\times \frac{\%\;\mathrm{nutrient}\;\mathrm{feces}}{\%\;\mathrm{nutrient}\;\mathrm{diet}}$$

### 2.6. Statistical Analysis

Zootechnical performance data are presented as the mean of three replicates’ standard deviation. Data were subjected to a one-way analysis of variance. When appropriate, means were compared by the Student–Newman–Keuls test. Before ANOVA, values expressed as percentages were subjected to arcsine square root transformation. Chi-square tests were used to compare associations between histological scores and fish diets [34]. Statistical significance was tested at a 0.05 probability level and calculated using the IBM SPSS Statistics software (version 21, Armonk, NY, USA).

Intestine histological traits were further associated with microbiota composition by correlating histology scores with genus relative abundance using the correlation function (name of function) in R [35] to find association patterns that could be modulated by diets. Considering the amino acid profile of each diet and the assessed feed intake throughout the trial, a possible association between this diet feature and microbiota composition and microbiome functionality was further investigated. Here, Pearson correlation between amino acid intake and microbiota relative abundance as well as the abundance of the identified MetaCyc pathways in each diet was explored, considering the ANIMAL diet as the reference for comparison. Both microbiota and microbiome pathways were previously filtered, and analysis included features that occurred in at least 30% of the samples. Correlations were considered significant when *p*-value < 0.01.

## 3. Results

### 3.1. Zootechnical Performance

Data on the apparent digestibility of experimental diets and zootechnical data after 46 days of feeding are presented in Table 2. The apparent digestibility coefficients (ADC) of protein and lipid were not affected by dietary treatments (*p* > 0.05). However, the energy digestibility of the INSECT diet was significantly higher than that of the PLANT diet (*p* < 0.05).

No mortality was recorded during the 46 days of experimental feeding. The final body weight (FBW) ranged between 66.4 and 73.0 g, which in the best-performing treatment represented a 6-fold increase in the initial body weight. The weight gain varied between 54.4 and 60.8 g. Fish fed the ANIMAL and BACTERIAL diets showed significantly higher FBWs and weight gain than those fed the PLANT and INSECT diets (*p* < 0.05). The feed conversion ratio (FCR) ranged between 1.00 and 1.13. Fish fed the PLANT and INSECT diets showed significantly higher FCRs than those fed the ANIMAL diet (*p* < 0.05). Feed intake was not significantly affected by dietary treatments (*p* > 0.05). Fish fed the ANIMAL and BACTERIAL diets showed significantly higher protein efficiency ratios (PER) than those fed the PLANT and INSECT diets (*p* < 0.05).

The whole-body composition of fish was not affected by the dietary treatments (data are presented in Appendix A). However, there were significant changes in whole-body nutrient retention (expressed as % of nutrient intake) (Table 3). Fish fed the ANIMAL diet showed significantly higher protein retention than those fed the INSECT diet (*p* < 0.05). Fish fed the ANIMAL diet showed a significantly higher lipid retention than those fed all other diets (*p* < 0.05), while fish fed the PLANT and BACTERIAL diets also showed a significantly higher lipid retention than those fed the INSECT diet (*p* < 0.05). Fish fed the ANIMAL diet showed a significantly higher energy retention than those fed all other diets (*p* < 0.05), while fish fed the BACTERIAL diet also showed a significantly higher energy retention than those fed the INSECT diet (*p* < 0.05).

### 3.2. Histology

Hindgut sections presented lymphocyte infiltrates within mucosa and lamina propria (LP). Additionally, mucosa presented a variable impact of intraepithelial degenerated forms compatible with degenerating enterocytes, as well as a variable presence of well-defined supranuclear vacuoles (SNV). LP presented with increased thickness, mostly affecting the distal end of intestinal folds. Intestinal submucosa (SM) presented with variable mixed cellular infiltrate, with a predominance of lymphocytes and eosinophilic granular cells (EGC) and a limited impact of congested blood vessels. The average scores for each group of main histopathological features are summarized in Table 4. Although a high total score (which reflects the sum of all histopathological features analyzed) was found in ANIMAL-fed fish, no difference was observed in the histopathological parameters analyzed. Average morphometric measurements and mucus counts are presented in Table 5. No statistical difference was found when also analyzing differential mucus cell counts, submucosa/mucosa ratio, and villi height.

### 3.3. Microbiome

#### 3.3.1. Descriptive Microbiome Results

A total of 36 samples were analyzed, resulting in 2,131,845 counts and 5248 features (OTUs). After filtering analysis and non-chimeric sequences were removed, a total of 2623 features were further analyzed up to the genus level. A DADA2 stats summary is presented in a Appendix A.

The first set of analyses examined the impact of diet on the modulation of BACTERIAL abundance in tilapia gut after feeding with diets containing different protein sources using classical microbiome analysis. The relative abundance of tilapia’s intestine at the phylum level shows that around 90% of the abundance comprised of the phyla Bacteroidetes, Fusobacteria, and Proteobacteria together (Appendix A). From these data, Proteobacteria resulted in the highest abundance (*p* < 0.05) in fish fed the ANIMAL diet (29.43%) when compared to other groups (14–16%). This phylum shows decreased abundance in the intestine when tilapia were fed diets containing alternative protein sources. When we analyze the abundance at the genus level, it can be seen that the greatest abundance by far (41–52%) comprises *Cetobacterium* in all treatments (Figure 2). A high abundance of genera (19 in total) from the Proteobacteria phylum can be observed, in addition to a great abundance of *Paludibacter* and Bacteriodales in the intestines of tilapia fed diets with different protein sources (PLANT, BACTERIAL, and INSECT).

This trend in genus differences was confirmed after DeSeq2 analysis. Figure 3 shows the genus that had differential abundance among groups. In this case, the genus *Paludibacter* was more abundant in the tilapia´s guts when fed BACTERIAL and PLANT diets when compared to other groups. The PLANT diet led to a higher abundance of *Novosphigobium* and *Laucobacter* when compared to other groups. On the other hand, *Cetobacterium* was lower in the ANIMAL-fed tilapia when compared to other groups.

Community richness was assessed using alpha diversity parameters, which show general quantitative measurements (e.g., Shannon and observed features), and quantitative measures incorporating phylogenetic relationships between the features (e.g., faith_pd) (Table 6). Despite the abundance differences shown previously, no statistical difference was observed in the richness of abundance in tilapia fed different diets after the Kruskal–Wallis test. The *p*-values of these tests are shown in the Appendix A (Appendix A).

Beta diversity instead represents the degree to which each sample differs from one another based on species abundance. The principal component analysis (PCA) plot shows (Figure 4) the overlapping of samples in all treatments and does not reveal differences in diversity or clear clusters grouping similar abundances. In addition, PERMANOVA pairwise statistical analysis did not reveal any statistical differences among groups for both weighted Unifrac (counting the abundance in each sample) and unweighted Unifrac (diversity of the samples without evaluating the abundance) (Appendix A).

When assessing the most correlated BACTERIAL genera and species with amino acid intake in each diet (Figure 5 and Figure 6), the majority of the BACTERIAL groups were groups with low abundance. The exceptions were *Pludibacter* sp., *Agrobacterium* sp., and *Propionibacterium* acnes. Some patterns were observed according to the diet the fish had. For example, *Pirellula* sp. abundance increased with the increase in intake of amino acids (Leu, Lys, Thr, Val, Met, Cys) but only in fish fed the INSECT diet. The same group showed increased abundance when fish that were fed the PLANT diet ingested less alanine and glycine. Other examples were *Flavisolibacter* sp., *Roseococcus* sp., and *Tatlockia* sp., which showed increased abundances when the intake of some amino acids was reduced but only in fish fed the PLANT diet. The latter was also modulated by the intake increase in non-essential amino acids Glx, Pro, and Ser in the PLANT diet.

Arginine and tyrosine intake seem to have a stronger effect on modulating microbiota in the fish fed diets with alternative proteins, and the PLANT and INSECT diets seem to have a higher impact on the modulation of the selected taxonomical groups.

#### 3.3.2. Prediction Analysis and Correlations

Among the pathways that were identified through the functional prediction analysis, we focused on those most directly related to amino acids and gut responses. These were amine and polyamine biosynthesis and degradation pathways, the amino acids biosynthesis and degradation pathways, and the fermentation pathways. Here, to assess a possible effect of amino acids in fish gut microbiota predicted functionality, a correlation was performed (Figure 7 and Figure 8) considering the AA intake from the different diets. Microbiota amine and polyamine biosynthesis pathways’ abundances were reduced when fish amino acid intake was higher in the INSECT diet. Amine and polyamine degradation pathways are modulated in a different manner depending on diets. Here, a reduction in Arg, His, and Lys intake in the PLANT diet group correlated with an increased abundance of the creatinine degradation pathway. On the other hand, increases in Leu, Tyr, and Glx intake are strongly related to a reduction in the abundance of the 4-aminobutanoate degradation pathway (Appendix A).

As expected, amino acid biosynthesis pathways’ abundances are modulated by alterations in amino acid intake and, interestingly, this seemed to happen more frequently in the BACTERIAL diet group (Figure 7 and Figure 8). A reduction in His intake relates to an increase in the biosynthesis of isoleucine, lysine, ornithine, and histidine biosynthesis pathways, but only in fish fed the BACTERIAL diet despite the same intake reduction being observed when fish are fed the INSECT diet. On the other hand, an increase in Met intake in fish fed the BACTERIAL diet relates to an increase in the isoleucine, lysine, ornithine, histidine, and tryptophane biosynthesis pathways’ abundances but also to a decrease in the methionine one. However, when the Met intake decreases in fish fed the INSECT diet, there is an increase in arginine and methionine biosynthesis and a strong reduction in the lysine biosynthesis pathway.

In the BACTERIAL diet, the intake of Thr was considerably higher when compared with the ANIMAL diet, and this relates to a reduction in amine and polyamine degradation and an increase in amino acid biosynthesis. The latter was loaded by an increase in the isoleucine, lysine, ornithine, histidine, and tryptophan biosynthesis pathways and a reduction in the methionine biosynthesis pathway. As seen, for instance, with the Met intake increase with alternative diets, the predicted response of the microbiome is strongly correlated in a differential manner depending on the core protein of the diet more than the amino acid alone. The same was observed regarding the predicted modulation of fermentation pathways. As an example, an increase in leucine uptake was correlated with an increase in the fermentation pathways responsible for fermenting pyruvate into butanoate and propanoate in fish fed the PLANT diet, whereas in fish fed the INSECT diet it was correlated with a reduction in glycerol degradation to butanol (Figure 7 and Appendix A).

## 4. Discussion

Nile tilapia feeds are generally characterized by moderate crude protein levels (28 to 36%), derived mostly from defatted plant oilseed meals and low levels of fishmeal and/or land-animal processed by-product meals (e.g., poultry meal). In line with the current drive towards the use of new sustainable and economically viable ingredients, to alleviate the dependency on fishmeal, several studies have assessed the use of emergent protein sources in Nile tilapia feeds [36]. Most studies on this topic have focused on the impact of such formulation changes in growth performance criteria, but very few have investigated the effects of protein sources on intestinal health criteria, such as microbiota or mucosal integrity. The relationship between dietary protein quality, namely variations in dietary amino acid supply, and the intestinal microbiota has gone largely unexplored. Emerging evidence suggests that dietary protein supply, both quantitative and qualitative, strongly impacts intestinal microbiota composition and function, and that protein–microbiota interactions can have critical impacts on host health [37]. In this context, the current study intends to contribute to a better understanding of the mechanisms underlying the modulation of gut morphology and microbiome in Nile tilapia juveniles fed with various dietary protein sources.

When trying to vary the dietary protein quality under practical formulations, we often rely on the use of ingredients of various origins. In our study, fish fed the commercial-like diet, with 30% of the total crude protein supply provided by fishmeal and poultry meal, served as the control (ANIMAL). While maintaining isoproteic levels among all diets, the 30% of the crude protein supply derived from ANIMAL protein sources was replaced by an equivalent protein supply derived from vegetables (PLANT), BACTERIAL biomasses (BACTERIAL), and insect meal (INSECT). The remaining 70% of the crude protein supply was kept fairly constant among all diets. The adoption of this formulation design tried to minimize changes in nutritional variables. A comparison of formulas differing in protein quality implies a trade-off between guaranteeing variable, but adequate, protein profiles (i.e., amino acid) without disregarding the other nutritional features of a given ingredient. For instance, the inclusion of insect meal, besides altering the amino acid profile, inevitably led to a higher dietary chitin level.

Overall, tilapia fed the ANIMAL and BACTERIAL diets showed significantly higher growth (FBW and weight gain) and protein efficiency ratios (PER) than those fed the PLANT and INSECT diets. Moreover, tilapia fed the PLANT and INSECT diets showed significantly higher FCRs than those fed the ANIMAL diet. While the whole-body composition of the fish was not affected by the dietary treatments, the tilapia fed the ANIMAL diet showed the highest whole-body retention of protein, lipids, and energy. Prior studies support our findings that BACTERIAL biomasses are adequate protein sources for tilapia, without any detrimental effects on zootechnical performance, including weight gain criteria [14]. In contrast to our findings, previous studies have shown that insect meals are good alternative protein sources in several fish species, including tilapia. The review of Aragão et al. [38] reports the use of insect meal as being controversial with some potential to disturb the gut epithelial barrier; however, no effect on growth performance paraments was reported. Sea bass (*D. labrax*) fed defatted *Tenebrio molitor* larvae replacing 50 and 100% of the fishmeal protein [39] and Nile tilapia (*O. niloticus*) fed the same levels of defatted black soldier (*Hermetia illucens*) larvae meal [40] presented no alterations in growth performance criteria. It is also worth pointing out that the differences found in the zootechnical performance criteria could not be directly associated with changes in the apparent digestibility of nutrients nor the histomorphological features analyzed in tilapia hindgut (e.g., villi length, villi height, mucus cell types), which were similar among dietary treatments.

The available literature data highlight the importance of microbiota in the health, performance, and various physiological functions of fish. The intestinal microbiota affects the nutritional metabolism, immunity, and disease resistance of the fish, while the host regulates the intestinal microbiota in a reciprocal way through both immune and non-immune factors. Criteria such as diet and environmental conditions are recognized to play a key role in the intestinal microbiota of fish and terrestrial animals [41,42]. At the phylum level, and for all dietary treatments, representatives of Bacteroidetes, Fusobacteria, and Proteobacteria were the main constituents of the BACTERIAL community, which is in agreement with previous studies on the characterization of intestinal microbiota in Nile tilapia [43,44,45,46]. The replacement of animal proteins (fishmeal and poultry meal) with the various alternative protein sources resulted in intestinal microbiota modulation at the phylum level. In comparison to PLANT, BACTERIAL, and INSECT treatments, fish fed the ANIMAL diet showed a lower abundance of Bacteroidetes and a higher abundance of Proteobacteria. This decrease in Proteobacteria abundance is an interesting feature because its swift increase in the gut has been associated with dysbiosis in mammals [47], and many genera of this group of bacteria can be pathogenic in fish [48]. Although assessed by a semi-quantitative approach, and without reaching a statistical significance, it is interesting to note that fish fed the ANIMAL diet presented the highest intestinal histopathological score for many of the parameters analyzed. However, we should remember that Proteobacteria are one of the main representatives of the commensal fish gut microbiome, and no definitive conclusions can be made on the impact of such change.

At the genus level, the greatest abundance (41–52%) comprised of *Cetobacterium* in all treatments. Although *Cetobacterium* abundance was lower in the fish fed the ANIMAL diet, this genus is commonly present in freshwater fish microbiota [43,46,49] and is known to have a high capacity to produce vitamin B12 and other compounds such as organic acids and enzymes [50]. In relation to the ANIMAL treatment, we observed a decrease in the abundance of several genus species within the Proteobacteria phylum and an increase in the genus *Paludibacter* and family *Bacteriodales* within the Bacteroidetes. The genus *Paludibacter* was more abundant in the guts of tilapia fed the BACTERIAL and PLANT diets when compared to other groups. *Paludibacter* sp. and other *Bacteroides* from the order Bacteroidales are anaerobic bacteria that have been associated with the degradation of carbohydrates (oligofructose) and production of propionate [7,51]. Tilapia fed diets supplemented with a carbohydrate-degrading enzyme, such as xylanase, presented an increase in *Paludibacter* in their intestinal tract [52]. Additionally, an increase in the genus *Paludibacter* was found in the intestinal tract of Senegalese sole with a 5% dietary supplementation of Ulva, a carbohydrate-rich macroalgae. The rise in *Paludibacter* in tilapia fed the PLANT diet may therefore be associated with a higher intake of certain carbohydrate fractions derived from soybean meal and corn gluten meal. However, this same hypothesis does not hold true for the enhanced *Paludibacter* levels found in tilapia fed the BACTERIAL diet. Nevertheless, a recent study with rainbow trout mentions that being fermentative bacteria, species from the genus *Paludibacter* can degrade dead BACTERIAL biomass and produce organic alcohols and fatty acid by-products [53]. Still, at the genus level, we found that PLANT-fed fish showed a higher abundance of *Novosphigobium* and *Leucobacter* when compared to all other groups. The *Novosphigobium* genus has generally been associated with polycyclic aromatic hydrocarbon (PAH)-degrading pathways, which does not seem to be a relevant variable in our trial. However, Xie et al. [54] suggest that *Novosphingobium* produces prolyl endopeptidase, which is an enzyme that degrades lignin present in plant substrates. The genus *Leucobacter* has mainly been associated with detoxifying pathways of chromium compounds. Although often detected in fish intestines and feces, information on its modulation by dietary factors is extremely scarce. Despite the changes found in microbiome abundance, no differences were found in a wide spectrum of diversity indexes in tilapia that were fed the various diets.

In this study, we chose to explore how variable amino acid (AA) intake impacts the intestinal BACTERIAL community in tilapia. It is known that AAs available in the intestine can be used by intestinal bacteria for their own protein synthesis and in catabolic reactions releasing numerous metabolites in the intestinal lumen, such as ammonia, hydrogen sulfide, polyamines, and phenolic and indolic compounds. In a recent review, Beaumont et al. [5] consider the effects of nourishing the gut microbiota with amino acids and explore the effects of the “aminobiotics”, a new term in functional nutrition. It is worth mentioning that in our study, apart from *Paludibacter* sp., *Agrobacterium* sp., *and Propionibacterium acnes*, the most correlated microbiota had low abundance. Lower-abundance taxa are normally not identified in core microbiomes; however, studies have shown a strong impact on the host’s health [49,55]. As an example, in invertebrate animals, such as termites, low-abundance microorganisms shift with dietary factors and drive the whole BACTERIAL community, leading to a higher resistance [50], whereas in humans several keystone low-abundance microorganisms are pathobionts associated with diseases [51]. How and to what extent fish gut microbiota utilize available AAs is still under study; however, in other species such as pigs, it is known that AAs in the intestine are the main constituents of ileum bacterial protein, with an unlikely AA de novo synthesis [6,7]. The utilization of AAs by intestinal bacteria is known to be dependent on species. This has been described in pigs [52,56,57], and it was evident in the present study where the modulation of AA intake had different outcomes in BACTERIAL taxon abundance. Although this may be due to many of the roles of AAs in the gut [8], the impact of AAs in taxon-specific metabolic pathway modulation might be a core feature and should be further addressed.

The *Paludibacter* genus was found to be a highly abundant member in the BACTERIAL community of the tilapia reported here and, as previously mentioned, its relative abundance increased when fish were fed diets with more BACTERIAL and PLANT proteins, the latter having a strong correlation with the higher intake of leucine and tyrosine. *Paludibacter* is considered to be primary fermentative bacterium that degrades BACTERIAL biomass and produces organic alcohols and intermediary products that can have beneficial effects in the gut (e.g., propionate, benzoate, acetate) [53]. Interestingly, when addressing the BACTERIAL pathways’ prediction, the increase in leucine intake in the PLANT protein-based diet was correlated with an increase in the pyruvate fermentation to butanoate and propanoate pathway. Whether this modulation is linked with the *Paludibacter* abundance shift is unknown; however, it indicates a possible relationship between AA intake and BACTERIAL modulation and function, with potential positive effects for the host that deserve further study. In contrast, the low-abundance genus *Pirellula* increased abundance in tilapia fed with all the alternative proteins, but only correlated with higher AA intake derived from INSECT protein in the diet, such as leucine, methionine, and cysteine. Here, the fermentation pathways were modulated as well; however, this was due to a reduction in the glycerol’s degradation to butanol. This exemplifies the intricacy of BACTERIAL utilization of accessible amino acids and suggests that the reactions are specific to each species and are probably correlated with the overall source of the ingredient, in addition to the availability of the AAs.

Considering the correlation found between microorganisms and AA intake, we observed that Arg, Tyr, Phe, and Ile show higher modulatory potentials. Tyrosine (Tyr) intake, for example, was higher in the PLANT group. It is interesting that despite Tyr normally being associated with inductions of DNA damage and lower mitochondrial respiratory capacity in intestinal epithelial cells [58], a positive correlation with fermentation pathways (pyruvate to butanoate and propanoate) was seen. This might be related to the strong positive correlation between Paludibacter abundance and the action of this taxon counteracting the possible negative effects derived from this in the PLANT diet. We also observed a higher input of Thr when fish were fed the BACTERIAL diet, and although no strong correlation was found with a taxonomic group, a reduction in 4-aminobutanoate degradation was predicted within the amine and polyamine degradation pathways. Threonine has been described as a gut barrier and immunity-enhancing amino acid [59] through the production of SCFA and other metabolites. Although dedicated research is required on this topic, it is inevitable to associate the observed good performance in the BACTERIAL group with this dramatic difference in Thr intake, which might be related to the several health-promoting functions already attributed to this AA [60]. On the other hand, Cys intake was lower in the BACTERIAL diet group. As for Thr, the different intake in Cys was not correlated with a specific BACTERIAL group; however, it was associated with a (predicted) reduction in microbiome 4-aminobutanoate degradation and the aromatic biogenic amine degradation pathways. Furthermore, 4-aminobutanoate is known for improving growth, digestive performance, and immunocompetence [5]. This effect is linked with a reduction in the detrimental effect of ammonia release in the epithelium [61,62], and might counteract the known negative effect of the Cys degradation by microbiota that releases hydrogen sulfide and potentiates inflammation [63]. Functional prediction of the microbiome is regarded as a valuable instrument for initiating novel perspectives on the impacts of nutritional manipulation in fish. Nevertheless, it is important to note that additional investigation is required to enhance and affirm these impacts, alongside the associations and their connection to a plausible cause-and-effect relationship involving the intake of amino acids, modulation of the microbiome, and the performance of the host.

## 5. Conclusions

Increasing knowledge about the impact of dietary protein quality on the composition of the gut’s BACTERIAL population is key for diet optimization toward performance. This study addressed the effect of variable dietary protein qualitative supplies (i.e., variations in amino acid intake patterns) on the intestinal microbiota of Nile tilapia. Here, we conclude that the core structure of the gut microbiota community of Nile Tilapia is stable to diet changes; however, it is at the genus level where we highlighted the shifts related to diet. To the best of our knowledge, this is the first study to correlate amino acid intake levels with microbiome modulation in the gut and with its functionality, the latter in predictive form. Lower-abundance taxa are the ones with a stronger correlation with amino acid intake, and although correlation does not mean causality, it suggests strongly that protein quality has a direct impact on these taxa, with potential implications for animal health. Variations in the protein qualitative supply have an impact on the functioning of the microbiota–host interface, and although further studies are needed to deepen the knowledge generated here, these results support future decisions on the protein consortiums to be used in Nile tilapia diets.

## Figures and Tables

**Figure 1 animals-14-00714-f001:**
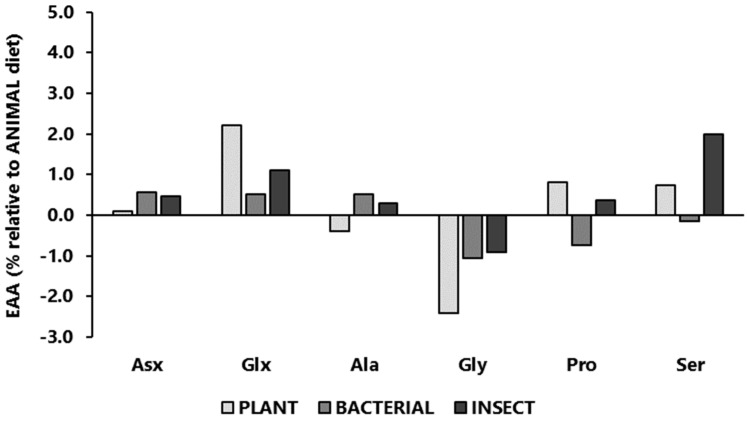
Intake of essential amino acids (EAAs; **above**) and non-essential amino acid (NEEA; **below**) profiles of experimental diets, expressed as a percentage relative to the amino acid composition of the ANIMAL diet.

**Figure 2 animals-14-00714-f002:**
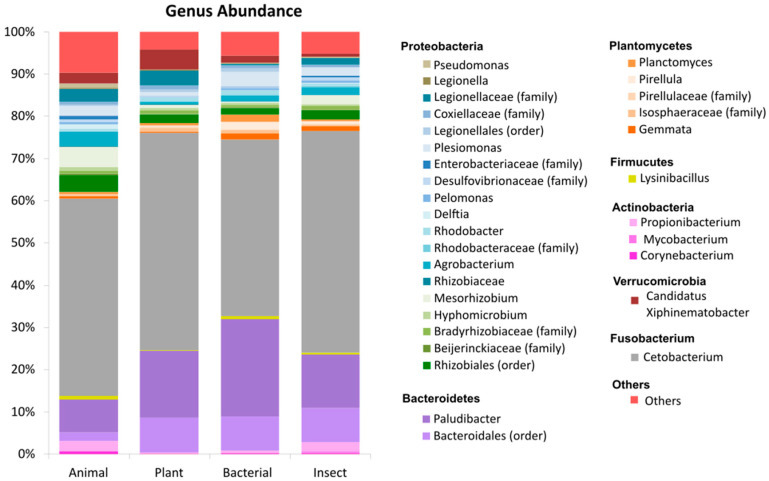
Percentage of relative abundance at the genus level for tilapia fed diets containing different protein sources. Legend represents the most abundant genus for each treatment. Others represent the genera with an abundance <0.3% in at least one treatment.

**Figure 3 animals-14-00714-f003:**
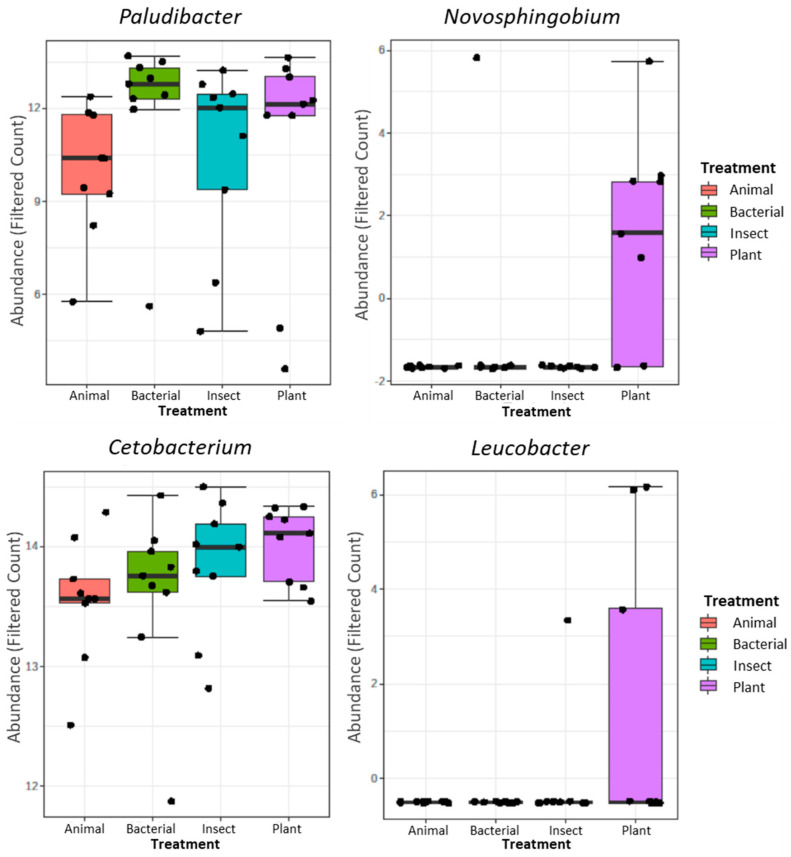
Genera with differential abundance among the experimental groups based on DeSeq2 (adjusted *p*-value < 0.05).

**Figure 4 animals-14-00714-f004:**
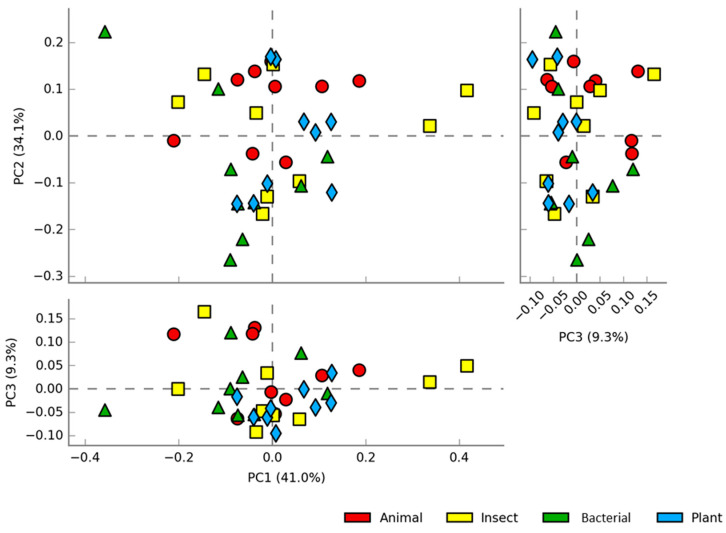
Score plots of the PCA at the genus level showing the distribution of abundance among samples of fish intestine fed diets containing different protein sources: ANIMAL, INSECT, PLANT, and BACTERIAL.

**Figure 5 animals-14-00714-f005:**
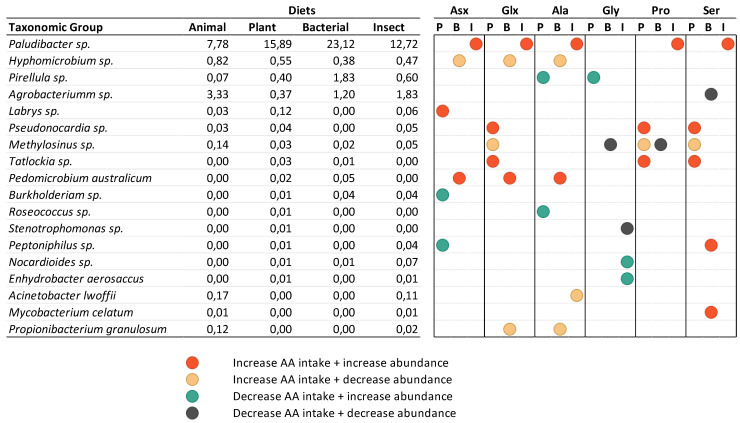
Correlation between non-essential amino acid (NEAA) intake and abundance of genera and species groups in different treatments in the intestine of tilapia (*O. niloticus*).

**Figure 6 animals-14-00714-f006:**
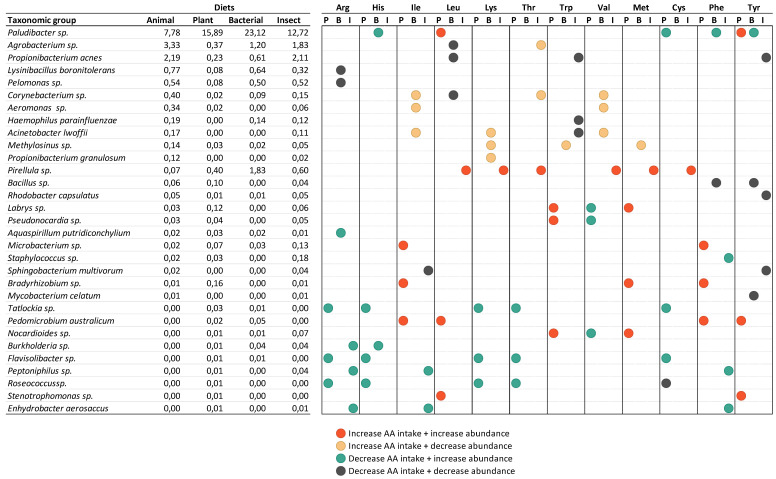
Correlation between essential amino acid (EAA) intake and abundance of genera and species groups in different treatments in the intestine of tilapia (*O. niloticus*).

**Figure 7 animals-14-00714-f007:**
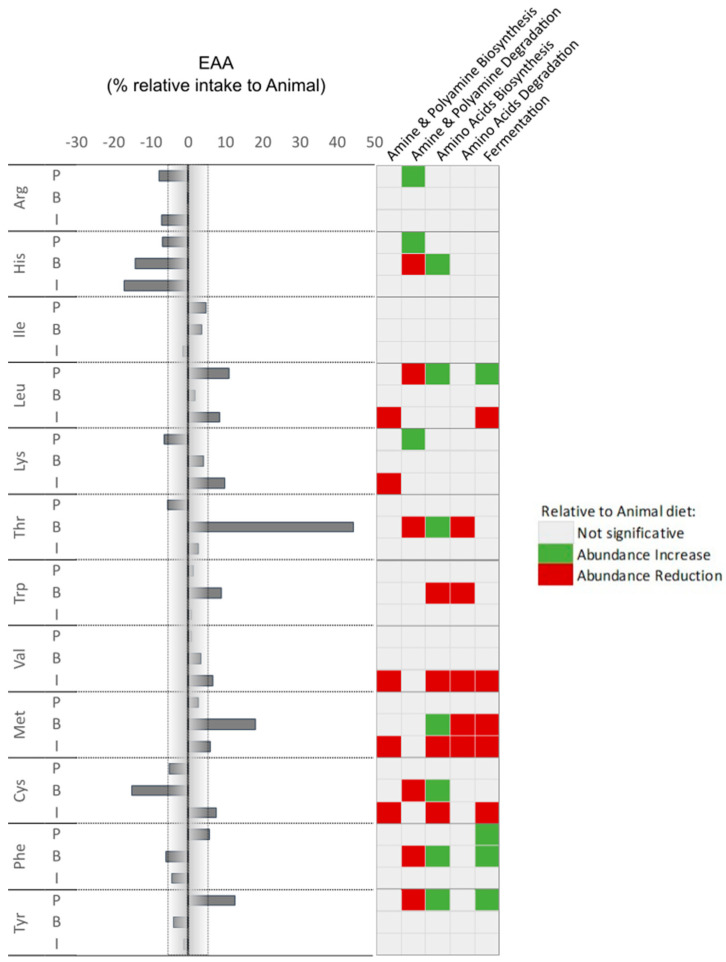
Percentage of essential amino acid (EAA) intake in correlation with an increase or reduction in prediction pathways relative to the ANIMAL diet.

**Figure 8 animals-14-00714-f008:**
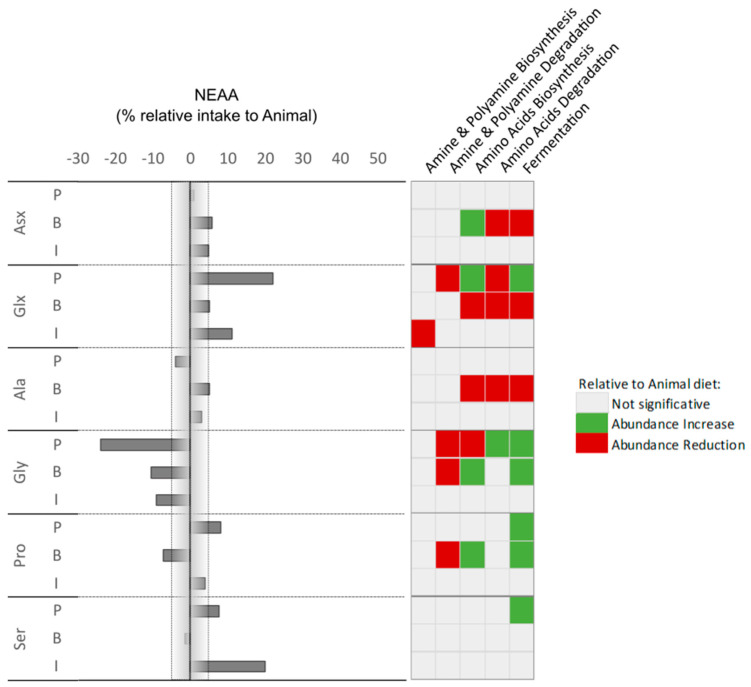
Percentage of non-essential amino acid (NEAA) intake in correlation with an increase or reduction in prediction pathways relative to the ANIMAL diet.

**Table 1 animals-14-00714-t001:** Ingredient composition and proximal analysis of the four diets tested.

Ingredients, %	ANIMAL	PLANT	BACTERIAL	INSECT
Fishmeal 60 ^1^	5.00			
Poultry meal ^2^	10.00			
Insect meal ^3^				13.00
Bacterial biomass (*C. glutamicum*) ^4^			6.50	
Bacterial biomass (*M. capsulatus*) ^5^			6.50	
Corn gluten meal ^6^	10.00	15.00	10.00	10.00
Soybean meal 48 ^7^	15.00	30.75	15.00	15.00
Rapeseed meal ^8^	15.00	15.00	15.00	15.00
Wheat bran ^9^	10.00	10.00	10.00	10.00
Rice bran ^10^	10.00	10.00	10.00	10.00
Wheat meal ^11^	6.30		5.65	6.60
Corn meal ^12^	10.00	7.75	10.00	10.00
Vitamin and mineral premix ^13^	1.00	1.00	1.00	1.00
Antioxidant ^14^	0.30	0.30	0.30	0.30
Monocalcium phosphate ^15^	1.10	2.60	2.40	2.50
Guar gum ^16^	0.50	0.50	0.50	0.50
L-Lysine ^17^	0.20	0.25	0.40	0.40
L-Threonine ^18^				0.10
DL-Methionine ^19^	0.10	0.15	0.25	0.20
Chromium oxide ^20^	1.00	1.00	1.00	1.00
Fish oil ^21^	1.00	1.40	1.40	1.40
Soybean oil ^22^	3.50	4.30	4.10	3.00
Proximate composition				
Moisture, %	7.3 ± 0.0	6.5 ± 0.4	6.3 ± 0.0	6.3 ± 0.0
Ash, %	6.9 ± 0.0	6.6 ± 0.0	6.6 ± 0.0	6.9 ± 0.0
Crude protein, %	35.0 ± 0.1	34.8 ± 0.2	34.5 ± 0.1	35.1 ± 0.1
Crude fat, %	9.0 ± 0.1	9.1 ± 0.1	8.9 ± 0.2	8.9 ± 0.1
Gross energy, MJ/kg	18.7 ± 0.0	18.8 ± 0.0	18.8 ± 0.1	18.8 ± 0.1
Chromium oxide, %	0.9 ± 0.0	0.9 ± 0.0	1.0 ± 0.0	0.8 ± 0.0

^1^ COFACO 60: 62.3% crude protein (CP), 8.4% crude fat (CF), COFACO, Portugal; ^2^ Poultry meal: 63% CP, 11% CF, SAVINOR UTS, Portugal; ^3^ Defatted meal worm powder (Tenebrio molitor) from ground larvae: 69% CP, 5.2% CF, AnimalPro Nutrition GmbH, Germany; ^4^ Aminopro NT70: 74% CP, 3% CF, MAZZOLENI SPA, Italy; ^5^ FeedKind: 71% CP, 8% CF, Calysta UK Ltd., United Kingdom; ^6^ Corn gluten meal: 58% CP, 4% CF, MPS, France; ^7^ Dehulled solvent-extracted soybean meal: 47% CP, 3% CF, CARGILL, Spain; ^8^ Defatted rapeseed meal: 34% CP, 2% CF, União Agrícola do Norte U.C.R.L., Portugal; ^9^ Wheat bran: 15.7% CP, 4.7% CF, Casa Lanchinha, Portugal; ^10^ Full-fat rice bran: 12.6% CP; 15.5% CF, Casa Lanchinha, Portugal; ^11^ Wheat meal: 10.2% CP; 1.2% CF, MOLISUR, Spain; ^12^ Corn meal: 10.1% CP; 4.2% CF, Casa Lanchinha, Portugal; ^13^ PREMIX Lda, Portugal: Vitamins (IU or mg/kg diet): DL-alpha tocopherol acetate, 100 mg; sodium menadione bisulphate, 25 mg; retinyl acetate, 20,000 IU; DL-cholecalciferol, 2000 IU; thiamin, 30 mg; riboflavin, 30 mg; pyridoxine, 20 mg; cyanocobalamin, 0.1 mg; nicotinic acid, 200 mg; folic acid, 15 mg; ascorbic acid, 500 mg; inositol, 500 mg; biotin, 3 mg; calcium pantothenate, 100 mg; choline chloride, 1000 mg, betaine, 500 mg. Minerals (g or mg/kg diet): copper sulfate, 9 mg; ferric sulfate, 6 mg; potassium iodide, 0.5 mg; manganese oxide, 9.6 mg; sodium selenite, 0.01 mg; zinc sulfate, 7.5 mg; sodium chloride, 400 mg; excipient wheat middling’s; ^14^ VERDILOX PX, KEMIN EUROPE NV, Heretals, Belgium; ^15^ MCP: 22.7% P, Premix Lda, Neiva, Portugal; ^16^ Seah International, Wimile, France; ^17^ Biolys: 54.6% Lysine, Evonik Nutrition & Care GmbH, Essen, Germany; ^18^ ThreAMINO: 98.5% Threonine, Evonik Nutrition & Care GmbH, Essen, Germany; ^19^ Rhodimet NP99, ADISSEO, Antony France; ^20^ Sigma-Aldrich, St. Louis, MO, USA; ^21^ Sopropêche, Wimile, France; ^22^ J.C. Coimbra Lda, Setubal, Portugal.

**Table 2 animals-14-00714-t002:** Apparent digestibility coefficients and zootechnical performance criteria.

	ANIMAL	PLANT	BACTERIAL	INSECT	*p*-Value
ADC Protein, %	86.4 ± 0.9	85.2 ± 0.5	85.8 ± 1.4	84.5 ± 1.9	0.401
ADC Lipid, %	91.7 ± 1.9	90.3 ± 2.8	91.9 ± 1.0	93.3 ± 1.0	0.325
ADC Energy, %	75.9 ± 2.2 ^ab^	71.5 ± 0.4 ^a^	74.4 ± 1.0 ^ab^	78.5 ± 2.7 ^b^	0.034
IBW (g)	12.3 ± 0.3	12.0 ± 0.2	11.9 ± 0.2	12.4 ± 0.5	
FBW (g)	73.0 ± 1.3 ^b^	66.4 ± 0.9 ^a^	72.7 ± 2.9 ^b^	67.2 ±1.4 ^a^	0.003
Weight Gain (g)	60.7 ± 1.4 ^b^	54.4 ± 1.1 ^a^	60.8 ± 2.9 ^b^	54.7 ± 1.1 ^a^	0.003
FCR	1.00 ± 0.03 ^a^	1.12 ± 0.07 ^b^	1.03 ± 0.04 ^ab^	1.13 ± 0.04 ^b^	0.026
Intake (%BW/day)	3.09 ± 0.08	3.39 ± 0.22	3.21 ± 0.10	3.37 ± 0.11	0.080
PER	2.86 ± 0.09 ^b^	2.56 ± 0.17 ^a^	2.82 ± 0.12 ^b^	2.53 ± 0.09 ^a^	0.018

Values are means ± standard deviation (n = 3). Different superscripts within a row denote statistical differences (*p* < 0.05).

**Table 3 animals-14-00714-t003:** Whole-body nutrient retention (expressed as % of nutrient intake).

Retention	ANIMAL	PLANT	BACTERIAL	INSECT	*p*-Value
Protein, %	42.2 ± 0.4 ^a^	37.5 ± 2.5 ^ab^	41.3 ± 2.3 ^ab^	36.7 ± 3.0 ^a^	0.046
Lipid, %	85.5 ± 0.7 ^c^	76.9 ± 3.5 ^b^	77.4 ± 1.5 ^b^	69.2 ± 1.4 ^a^	<0.001
Energy, %	36.6 ± 0.7 ^c^	30.2 ± 1.8 ^ab^	31.8 ± 0.8 ^b^	28.5 ± 1.4 ^a^	<0.001

Values are means ± standard deviation (n = 3). Different superscripts within a row denote statistical differences (*p* < 0.05).

**Table 4 animals-14-00714-t004:** Histopathological features analyzed in tilapia hindgut fed diets containing different protein sources: ANIMAL, BACTERIAL, INSECT, and PLANT (mean ± standard deviation, n = 3).

Diet	Lamina Propria		Mucosa			Sub Mucosa	
	Infiltrate	Increased Thickness	Degenerative Forms	Reduction in SNV	Infiltrate	Infiltrate	Congestion
ANIMAL	1.25 ± 0.89	1.44 ± 0.5	1.00 ± 0.71	1.69 ± 0.7	1.94 ± 0.82	2.13 ± 0.44	0.25 ± 0.71
BACTERIAL	0.67 ± 0.43	0.78 ± 0.51	1.00 ± 0.75	0.89 ± 0.78	1.11 ± 0.6	1.22 ± 0.57	0.00 ± 0.00
INSECT	0.67 ± 0.52	0.67 ± 0.61	1.00 ± 0.89	0.33 ± 0.52	1.25 ± 0.52	1.5 ± 0.63	0.00 ± 0.00
PLANT	0.38 ± 0.52	0.56 ± 0.62	0.63 ± 0.58	1.00 ± 0.93	1.21 ± 0.59	1.44 ± 0.62	0.38 ± 0.52

**Table 5 animals-14-00714-t005:** Histomorphology features analyzed in tilapia hindgut fed diets containing different protein sources: ANIMAL, BACTERIAL, INSECT, and PLANT (mean ± standard deviation, n = 3).

Diet	Diameter (mm)	Villi (mm)		Mucus Cells		
	Ratio S/M	Average Height	Average Width	% Neutral Mucins	% Acid Mucins	% Mixed Mucin
ANIMAL	4.03 ± 2.06	0.27 ± 0.03	0.17 ± 0.06	8.38 ± 10.38	76.98 ± 18.5	14.65 ± 12.14
BACTERIAL	4.55 ± 3.68	0.29 ± 0.11	0.15 ± 0.05	13.18 ± 9.58	72.48 ± 21.98	14.33 ± 13.11
INSECT	5.98 ± 7.26	0.32 ± 0.07	0.17 ± 0.06	13.88 ± 10.59	66.94 ± 16.23	19.19 ± 11.55
PLANT	8.99 ± 8.71	0.29 ± 0.09	0.14 ± 0.03	15.72 ± 14.59	68.22 ± 17.48	16.06 ± 13.41

**Table 6 animals-14-00714-t006:** Alpha diversity parameters results for Shannon, faith_pd, and observed features for tilapia fed diets containing different protein sources: ANIMAL, INSECT, PLANT, and BACTERIAL.

Treatments	Shannon	faith_pd	Observed Features
ANIMAL	5.64 ± 0.32	12.98 ± 4.39	169.44 ± 52.53
BACTERIAL	5.35 ± 0.82	11.63 ± 8.77	149.44 ± 88.20
INSECT	5.31 ± 0.87	15.51 ± 10.08	155.88 ± 94.32
PLANT	5.38 ± 0.42	13.55 ± 4.38	177.22 ± 55.62

## Data Availability

Data are contained within the article.

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
