# Peer review of "Dietary Protein Quality Affects the Interplay between Gut Microbiota and Host Performance in Nile Tilapia"

_animals, 2024, doi:10.3390/ani14050714_

Round 1
Reviewer 1 Report
Comments and Suggestions for Authors
Dietary Protein
The authors need to revise the manuscript to make it more understandable to English readers.
The title needs to be rewritten to reflect that all diets contain plant protein meals. The conclusions need to reflect this and the authors need to state how keeping the plant protein ingredients int he diet affected their analysis and interpretation. Conclusions need to be based on ingredient inclusion level and not broadly on ingredient type.
Table 1: Why did the authors increase corn gluten meal and soy bean meal in the plant diet and not the other ingredients?
Line 210-217: Were the three samples pooled or analyzed individually? Please clarify this in the methods as they are individually analyzed.
Table 2: Either add SGR to the discussion or remove it from the manuscript.
Table 4: Why would the animal diet have an overall higher score compared to the other diets.
Line 517: They authors need to relate FCR, SGR, PER to other studies and not just restate the results. They seem to be similar to a cited study but how are they similar? The authors also need to include dietary inclusion rates for the various ingredients as that will affect the results.
Conclusions: They authors had only 1 level for each experimental diet and need to frame the discussion and conclusions based on the single level. They need to keep the conclusions tailored to the study design and not keep them as broad statements.
Comments on the Quality of English LanguageSee above. There in sentences throughout the manuscript that are hard to understand.
Author Response
The authors thank the reviewer´s comments. Please find below our changes and ideas. Regards,
The authors need to revise the manuscript to make it more understandable to English readers.
A: The manuscript was carefully revised and English writing was analysed. Changes were made throughout the text. Please check changes in red colour.
The title needs to be rewritten to reflect that all diets contain plant protein meals. The conclusions need to reflect this and the authors need to state how keeping the plant protein ingredients int he diet affected their analysis and interpretation. Conclusions need to be based on ingredient inclusion level and not broadly on ingredient type.
A: We acknowledge the reviewer’s comment and modified the title. The term protein “source” has been replaced by protein “quality”. This change was made also on several other sections of the manuscript. The study aimed to assess the effects of changing the dietary protein profile on the various gut-related criteria. The main variable among the experimental diets was not the ingredients itself, but the nature of 30% the total protein supply, which was derived from either fishmeal and poultry meal (ANIMAL), dehulled solvent extracted soybean meal and corn gluten meal (PLANT), microbial biomasses (MICROBIAL) or a defatted insect meal (INSECT). The remaining 70% of the total protein supply was derived from plant ingredients (corn gluten meal, dehulled solvent extracted soybean meal, rapeseed meal, wheat gran, full-fat rice bran, wheat, and corn meal) and were kept relatively constant among the various diets. This approach ensures that formulas would reflect an industrial applicability, since for instance, a total elimination of plant ingredients would be totally non-aligned with commercial formulas for Nile tilapia. However, we believe that the 30/70 concept of the dietary protein supply allows a controlled and relevant change on the dietary protein profile of the various diets.
Table 1: Why did the authors increase corn gluten meal and soy bean meal in the plant diet and not the other ingredients?
A: As mentioned before, our variable among the experimental diets were not the ingredients itself, but the nature of 30% the total protein supply. In the PLANT diet, we have chosen to raise solely the corn gluten meal (CGM) and soybean meal (SBM) for two reasons. First, these are often the most relevant protein sources in tilapia formulas. Secondly, they both possess relatively high crude protein levels (in our case 58% for CGM and 47% for SBM) which makes them more comparable with the animal, microbial and insect meals used in the other diets. Raising all plant ingredients in the PLANT diet, some of them with very low protein contents (12-15% in wheat and rice brans) would result in drastic formulation changes, that would compromise an effective comparison of the targeted change (30%) of the protein supply among diets.
Line 210-217: Were the three samples pooled or analyzed individually? Please clarify this in the methods as they are individually analyzed.
A: The 36 samples were individually analysed. The text on line 210 was modified to better understanding of the protocol.
Table 2: Either add SGR to the discussion or remove it from the manuscript.
A: The authors thank for the question. We have changed the discussion to add more data of SGR. Even though, there were important differences in the zootechnical parameters, the focus of this manuscript and the discussion was the differences in microbiome, therefore further effort in discuss this theme was made in the discussion.
Table 4: Why would the animal diet have an overall higher score compared to the other diets.
A: Total score from histology samples reflects the sum of scores of all semiquantitative histopathological measurements analysed. ANIMAL diet presents a higher histopathological score than other diets, however, with no significant difference from the other groups. We would like to highlight that the average score (considering all parameters) for ANIMAL was 1.2 and for others was 0.8 and both reflect mild incidence of histopathological features, therefore not reflecting more damage. The authors decided to remove the total score column from the manuscript to avoid misunderstandings from readers.
Line 517: They authors need to relate FCR, SGR, PER to other studies and not just restate the results. They seem to be similar to a cited study but how are they similar? The authors also need to include dietary inclusion rates for the various ingredients as that will affect the results.
A: The objective of this study was to evaluate the intestinal microbiota dynamics with different protein sources inclusion. The authors decided to reduce the discussion of growth paraments because this was not the objective of this study. However, we did mentioned some results of growth parameters indicators such as FCR, SGR and PER because they presented some differences and it might me relevant for readers. However. Part of the discussion about these parameters was added to the text.
Conclusions: They authors had only 1 level for each experimental diet and need to frame the discussion and conclusions based on the single level. They need to keep the conclusions tailored to the study design and not keep them as broad statements.
A: The study was not aiming the evaluation of an ingredient in particular, therefore formulas were not produced considering different inclusion levels of the experimental ingredients. The study design was focused on changing a block of the protein bulk inclusion. That means we have to work with blocks and indeed we used only one level per block. We don’t think that this is a drawback since it keeps the study with a high level of industrial applicability, and it allows us to answer our question: Is the intestinal microbiota being modulated by different protein sources? The conclusion was changed for better reading.
Reviewer 2 Report
Comments and Suggestions for Authors
Dear authors,
As requested, I reviewed the manuscript “Dietary protein source affects the interplay between gut microbiota and host performance in Nile tilapia” by do Vale Pereira G, Teixeira C, Couto J, Dias J, Rema P, Goncalves AT. The work focused on the effects on different gut parameters (especially the crosstalk between dietary source and gut microbiota) on Nile Tilapia administered with 4 diets different by protein (amino acids) source for 46 days. The work showed that different protein sources can interfere directly with the crosstalk of bacteria pathways and amino acid intake.
The paper is really interesting and well developed. A robust analysis is included with promising results for further studies. However, the work shows some and, for this, I suggest to accept the paper with minor revisions, which are as follows:
1. The Introduction is well developed, but poor in references. The authors should expand the list.
2. In Materials and Methods, in line 96, the authors should clarify what EPA and DHA mean.
3. The authors should clarify why in the growth trial fish have been raised in 90 L tanks with no acclimatization period and fed three times per day whereas in the second trial fish have been raised in 60 L tanks with 16 days of acclimatization period and fed twice a day. Furthermore, the authors should state the number of fish included in the original stock.
4. Some references in the text cannot be found in the list at the end of the paper (e.g., Bolin et al., 1952; AOAC, 2006; ISO, 2016). Furthermore, the references should be formatted using a single style.
5. In Materials and Methods, in “2.4.3. Histology”, the authors say that this analysis has been performed also for the liver but this is not mentioned either in the sampling or in the results.
6. More detailed information about the protocol used for mucus cell counts should be added.
7. In Supplementary Materials, in Table S2 the units of measure need to be added.
8. The authors might add the pictures of histological sections analyzed in “3.2. Histology”. This might help the reader in the comprehension of the criterion used by the authors in the observations of the histopathological features.
9. In Table 4, how has total score been calculated? Furthermore, in Table 5 the ratio S/M should not have any units of measure.
10. Finally, the authors might improve the conclusions adding some more detailed information and /or observations.
Thank you very much for your attention to my opinion.
Author Response
The authors thank the reviewer´s comments. Please find below our comments and ideas.
As requested, I reviewed the manuscript “Dietary protein source affects the interplay between gut microbiota and host performance in Nile tilapia” by do Vale Pereira G, Teixeira C, Couto J, Dias J, Rema P, Goncalves AT. The work focused on the effects on different gut parameters (especially the crosstalk between dietary source and gut microbiota) on Nile Tilapia administered with 4 diets different by protein (amino acids) source for 46 days. The work showed that different protein sources can interfere directly with the crosstalk of bacteria pathways and amino acid intake.
The paper is really interesting and well developed. A robust analysis is included with promising results for further studies. However, the work shows some and, for this, I suggest to accept the paper with minor revisions, which are as follows:
- The Introduction is well developed, but poor in references. The authors should expand the list.
A: The authors understand the comment and thank for it. The introduction was addressed in a way to focus on the amino acid interaction with gut health and microbiome. There is a lack of research done in this sense and consequently even less literature available. However, we made an extra effort and added some references in the introduction as suggested.
- In Materials and Methods, in line 96, the authors should clarify what EPA and DHA mean.
A: the changes were made on the text.
- The authors should clarify why in the growth trial fish have been raised in 90 L tanks with no acclimatization period and fed three times per day whereas in the second trial fish have been raised in 60 L tanks with 16 days of acclimatization period and fed twice a day. Furthermore, the authors should state the number of fish included in the original stock.
A: That is the standard protocol for Apparent digestibility measurements. The feed is offered tw times a day to have a better control of the feces collection and avoid the collection
- Some references in the text cannot be found in the list at the end of the paper (e.g., Bolin et al., 1952; AOAC, 2006; ISO, 2016). Furthermore, the references should be formatted using a single style.
A: Thank you for the notice. The references were added in the reference list and cited in the main text.
- In Materials and Methods, in “2.4.3. Histology”, the authors say that this analysis has been performed also for the liver but this is not mentioned either in the sampling or in the results.
A: Thank you for the comment. That was a mistake. We removed the word liver in the text.
- More detailed information about the protocol used for mucus cell counts should be added.
A: As this service was made by an external company we added the protocol they shared with us.
- In Supplementary Materials, in Table S2 the units of measure need to be added.
A: Thank you for the request, the changes were made as below.
Table S2: Whole body composition of fish under the dietary treatments.
Diet |
Fresh matter |
||||
Moisture (%) |
Ash (%) |
Protein (%) |
Fat (%) |
Energy (kJ/g) |
|
Initial |
76,28 |
4,26 |
14,99 |
4,07 |
4,88 |
ANIMAL |
73,93±0,42 |
3,12±0,18 |
14,80±0,34 |
6,99±0,23 |
6,07±0,07 |
PLANT |
73,90±0,85 |
3,16±0,03 |
14,70±0,63 |
7,18±0,12 |
6,12±0,18 |
MICROBIAL |
73,99±0,46 |
3,61±0,35 |
14,67±0,44 |
6,61±0,23 |
5,94±0,12 |
INSECT |
74,53±0,63 |
3,16±0,19 |
14,60±0,64 |
6,44±0,13 |
5,82±0,17 |
- The authors might add the pictures of histological sections analyzed in “3.2. Histology”. This might help the reader in the comprehension of the criterion used by the authors in the observations of the histopathological features.
A: Thank you for the comment. We agree that adding pictures give the readers more visual information about the results, however the authors decided to not add pictures of the histology because there were no statistical differences.
- In Table 4, how has total score been calculated? Furthermore, in Table 5 the ratio S/M should not have any units of measure.
A: We appreciate this comment. The overall score is calculated by summing all scores given in all histopathological measurements analysed. We decided to show the 7 most relevant results for this work. However, the authors decided to remove this column from the results to avoid misunderstandings.
- Finally, the authors might improve the conclusions adding some more detailed information and /or observations.
A: We acknowledge this comment. The Conclusion was reviewed and changed as requested.
Round 2
Reviewer 1 Report
Comments and Suggestions for Authors
The English still needs improvement. What additional information is gained from including specific growth rate? The authors do not compare it to SGR from other studies (this would not be appropriate) and the same information is gained from final weight as all the fish started with the same mass. They mention it in the discussion but it does not add to the manuscript. Also, fish do not grow exponentially and this is a poor model for fish growth.
The authors mention AA intake but need to present AA intake data to support their conclusions.
Do the authors have amino acid digestibility data to support AA intake?
Line 13. This sentence needs to be rewritten.
Line 17: Do the authors mean interfere or affects? Are the cross-talk bacterial pathways modified or do the pathways not function?
Line 23: Change Microbial to Bacterial to avoid confusion. Microbial can mean fungus, yeast, microalgae and bacteria.
Comments on the Quality of English LanguageSee above
Author Response
The English still needs improvement. What additional information is gained from including specific growth rate? The authors do not compare it to SGR from other studies (this would not be appropriate) and the same information is gained from final weight as all the fish started with the same mass. They mention it in the discussion but it does not add to the manuscript. Also, fish do not grow exponentially and this is a poor model for fish growth.
A: To accommodate the reviewer’s comment, we have decided to replace the SGR by a more simplistic criteria absolute weight gain (final body weight – initial body weight). The manuscript text has been adapted accordingly.
The authors mention AA intake but need to present AA intake data to support their conclusions.
A: Absolute values for amino acid intake per treatment are presented in Table S1 Supplementary material Appendix A). In the manuscript, the AA intake data are presented as Figure1, but expressed as percentage relative to the amino acid intake of the ANIMAL treatment, which being a more typical formula for Tilapia served as control for subsequent analysis (Figures 7 and 8).
Do the authors have amino acid digestibility data to support AA intake?
A: Unfortunately, we only have data on the digestibility of protein, lipids and energy.
Line 13. This sentence needs to be rewritten.
A: L13-L16 was revised accordingly for clarity. “: Gut health depends on a complex network interaction between the host and the mucosal-associated microbiota and it is modulated by dietary inputs. To gain knowledge on this interaction and shed light on the nutritional impacts at this level, this study assessed the effect of dietary protein quality on the modulation of different gut health parameters.”
Line 17: Do the authors mean interfere or affects? Are the cross-talk bacterial pathways modified or do the pathways not function?
A: We understand the confusion, and we revised the sentence for clarity. “Results indicate that different protein quality modulates the relationship between bacteria functional pathways and amino acid intake.” We revised for modulate since there was no inferred malfunction or deleterious effect.
Line 23: Change Microbial to Bacterial to avoid confusion. Microbial can mean fungus, yeast, microalgae and bacteria.
A: Following the reviewer’s suggestion, the terminology “Microbial” was modified to “Bacterial” throughout the manuscript including figures and tables.
